# Analysis of Endodontic Successes and Failures in the Removal of Fractured Endodontic Instruments during Retreatment: A Systematic Review, Meta-Analysis, and Trial Sequential Analysis

**DOI:** 10.3390/healthcare12141390

**Published:** 2024-07-11

**Authors:** Mario Dioguardi, Corrado Dello Russo, Filippo Scarano, Fariba Esperouz, Andrea Ballini, Diego Sovereto, Mario Alovisi, Angelo Martella, Lorenzo Lo Muzio

**Affiliations:** 1Department of Clinical and Experimental Medicine, University of Foggia, Via Rovelli 50, 71122 Foggia, Italy; corrado_dellorusso.564493@unifg.it (C.D.R.); filippo.scarano@unifg.it (F.S.); fariba_esperouz.560248@unifg.it (F.E.); andrea.ballini@unifg.it (A.B.); diego_sovereto.546709@unifg.it (D.S.); lorenzo.lomuzio@unifg.it (L.L.M.); 2Department of Surgical Sciences, Dental School, University of Turin, 10127 Turin, Italy; mario.alovisi@unito.it; 3DataLab, Department of Engineering for Innovation, University of Salento, 73100 Lecce, Italy; angelo.martella@unisalento.it

**Keywords:** endodontic, fractured, retreatment, ultrasonic, separated, broken, failure endodontic, instrument endodontic

## Abstract

This study presents a systematic review with meta-analysis to evaluate the success rates of endodontic retreatments in teeth where separated instruments are located within the roots. The search and selection of studies were conducted across two databases, SCOPUS and PubMed, as well as the Cochrane Library registry, yielding a total of 1620 records. Following the selection process, 11 studies were included in the systematic review. Overall, out of 1133 retreated teeth, there were 172 failures in instrument removal and 55 perforations. The meta-analysis results indicate that failures are more frequent when instruments are located in the apical third, with a failure rate of 21%, compared to an 8.8% failure rate in the middle/coronal third. The anatomy of the root canals, particularly the location of the separated instruments, significantly influences the success rates.

## 1. Introduction

One of the challenges in endodontics is dealing with dental elements containing fractured instruments within the canals. The fracture rate of endodontic instruments is estimated to range from 1.83% to 8.2% [1], depending on the type of instrument used. Indeed, a retrospective study conducted in 2024 by Alamoudi et al. [1] identified 108 cases of separated instruments out of 3150 teeth, with molars accounting for 96 percent of cases. The incidence of fracture in molars compared to premolars is approximately 2.9 times higher, as reported by Iqbal et al. in 2006 [2]. Additionally, Suter et al. in 2005 reported that half of the fractured instruments were located in the mesial roots of lower molars in their retrospective study [3], consistent with Nevares et al. in 2012 [4].

As a result, in clinical practice, most dentists have unfortunately witnessed this event at least once. The perceptions and clinical modalities in which an instrument separation occurs are closely linked to the nature of the instrument itself. Nickel-titanium (NiTi) instruments exhibit greater elastic flexibility and torsional fracture resistance compared to stainless steel (SS) instruments. Fractures of SS instruments are often preceded by clinically visible distortions, while NiTi rotary files may fracture without warning, as the distortion is not always visible. NiTi rotary instruments are prone to separation due to cyclic flexural fatigue, torsional yielding, or a combination of both [5].

The presence of fractured instruments within root canals has adverse effects on the prognosis of the affected tooth. Studies have demonstrated how the presence of fractured instruments within the canals influences the disinfection phase of the canals and the sealing phase of the apical foramen [6]. Therefore, in the presence of canal contamination at the time of separation or apical pathology, it is necessary to evaluate whether the non-bypassable fragment impedes the decontamination of the root canal, considering removal.

It is logical to distinguish between teeth treated before the development of periapical lesions and those treated in the acute phase without pulp necrosis. This distinction is necessary because teeth with periapical lesions have an 85.4% healing rate following root canal therapy, compared to teeth without lesions where the success rate increases to 94.6% [7]. The presence of fractured instruments within the canals and a periapical lesion further reduces the success rate.

Consequently, instrument removal is always recommended, starting with conservative treatment and, in specific cases, resorting to periapical surgical treatment [8].

Various conservative techniques are employed in endodontics, with the most commonly used and described in the literature being the technique that utilizes a combination and sequential use of Gates Glidden burs and ultrasonic tips, as detailed by Ruddle [9]. Additionally, bypass techniques are employed, which do not involve the removal of the instrument but rather its traversal to incorporate it into the root canal filling material.

Previous literature reviews on separated instruments have highlighted the lack of guidelines for managing and treating teeth with separated instruments within the endodontic space. Indeed, a review conducted by Madarati et al. [10] indicates that clinical experience, upon which balanced decision-making relies, must consider evaluations of factors such as the type of root on which the instrument separated, the stage of treatment in which the root canal containing the separated instrument is found, the skill of the endodontist, the available resources (intraoperative microscope and ultrasonics for instrument removal), the presence of periapical lesions, the strategic role of the tooth in broader rehabilitation, and potential complications.

Conversely, McGuigan et al. focused on the incidence of instrument separation and the types of instruments, reporting that the incidence of instrument separation in NiTi instruments is similar to that in stainless steel (SS) files [5].

Despite the technological advancements in endodontics and materials, the presence of separated instruments within the canals remains a primary challenge for practitioners involved in endodontic retreatments. The current medical scientific literature offers various guidelines on the management and outcomes of endodontic retreatment; however, there is a notable gap concerning the specific success and failure rates associated with teeth containing separated instruments. While some studies have addressed the general outcomes of retreatment, comprehensive analyses focusing on the impact of separated instruments are limited.

This gap in the literature underscores the need for a systematic review and meta-analysis to aggregate existing data and provide a clearer understanding of the prognostic implications of separated instruments in endodontic retreatment. By synthesizing and aggregating the findings of multiple studies, this research aims to offer valuable insights into the factors influencing success and failure rates, thereby guiding clinical decision-making and improving patient outcomes.

This study aims to conduct a systematic literature review to evaluate the failure rate of endodontic retreatments that involved a separated instrument within the root canal prior to retreatment. The reviewers established primary criteria for the selection and inclusion of studies to address the objectives of the systematic review, based on several factors. Studies were included and considered in the review if they evaluated the outcomes of endodontic retreatment where separated instruments were present; provided sufficient data on success and failure rates, including the specific locations of separated instruments within the root canals; used consistent and comparable criteria for defining the success and failure of endodontic treatment; and were published in peer-reviewed medical scientific journals and available in English. These criteria were chosen to ensure that the meta-analysis synthesized relevant and high-quality data, allowing for a thorough analysis of the impact of separated instruments on the success rates of retreatment.

The justification for conducting this systematic review stems from the fundamental need to enhance the predictability and success of endodontic retreatment procedures. By providing valuable insights, this review aims to contribute to the development of more effective treatment strategies.

## 2. Materials and Methods

### 2.1. Protocol and Registration

The systematic review was written following the PRISMA guidelines (Preferred Reporting Items for Systematic Reviews and Meta-Analyses) [11]. All search procedures, selection, and data extraction followed the Cochrane Handbook guidelines, and the review protocol was submitted and registered on the PROSPERO platform with the registration number CRD42024546246.

### 2.2. Eligibility Criteria

All clinical studies pertaining to the removal of separated endodontic instruments in the roots were considered potentially eligible, provided they reported data specifically on the success or failure of instrument removal and the success or failure of endodontic retreatment.

The formulated PICO question was as follows: What is the failure rate of endodontic retreatments involving a separated instrument within the roots prior to retreatment?

(P) Participants: patients with teeth containing separated instruments within the roots.

(I) Intervention: endodontic retreatment with removal of the separated instrument.

(C) Control: patients with teeth without separated instruments within the roots.

(O) Outcome: removal of the instrument or healing of associated endodontic lesions.

The inclusion criteria were as follows: clinical studies (retrospective, case-control, randomized trials, and prospective) reporting data on the number of failures or successes of teeth retreated endodontically and involving separated endodontic instruments within the roots.

Studies were excluded from the review if they had any of the following criteria: a high dropout rate: studies in which more than 20% of initial participants dropped out of the study before its conclusion; this criterion was chosen because a high dropout rate may compromise the validity and reliability of the study results; insufficient data: studies that did not provide sufficient data on the success and failure rates of endodontic retreatment, including details on the location of individual instruments within root canals; inconsistent definitions of success and failure: studies that used non-standardized or non-comparable criteria to define success and failure of endodontic treatment.

Furthermore, the following exclusion criteria were applied: all reports related to systematic reviews, scoping reviews, mapping reviews, narrative reviews, case reports, case series with a low number of included cases, in vitro and in silico studies, studies not reporting data on the failures or successes of retreatment or where it was not possible to extract data regarding teeth containing separated instruments or their eventual removal, studies published in a language other than English, those lacking an abstract in English, and studies deemed to be at high risk of bias.

### 2.3. Sources of Information, Research and Selection

The search for articles and reports was conducted using online search engines by two reviewers, who are also the authors of the manuscript (M.D. and C.D.R.). Preliminary exclusion criteria included linguistic restrictions: reports without at least an abstract in English were excluded using automated tools available in the databases. The search engines and databases utilized were PubMed, Scopus, and the Cochrane Library. Additionally, a grey literature search was conducted using Google Scholar, Science Direct, and Open Gray. To further minimize publication bias, the references of previous reviews on the removal of separated instruments were examined. The search, including the most recent update of the identified records, was completed on 4 April 2024.

The search terms were chosen to encompass the various terminologies used in the literature for endodontic retreatment and the presence of separated instruments. The terms were combined using Boolean operators to ensure comprehensive coverage of relevant studies.

For the database search, the following terms were used in combination: broken instrument endodontic, separated instrument endodontic.

Terms such as “divorce” not relevant to endodontic and medical themes were not chosen by the authors as keywords but were generated by PubMed. Nevertheless, they are reported in the manuscript by the authors for the reproducibility of the research and for correct consistency with the number of records identified by PubMed.

The following search terms were used on PubMed:

Search: (separated instrument endodontic) OR (broken instrument endodontic) OR (Ultrasonic technique separated instrument) Sort by: Most Recent; The records generated regarding “divorce” were excluded from the selection.

On the Scopus and Cochrane Library platforms, a search was conducted in these two databases for the presence of the terms (broken OR separated) AND endodontic in the title, abstract, and keywords (TITLE-ABS-KEY) (Table 1).

The identified records were imported into EndNote, and duplicates were automatically detected and removed using the software’s “find duplicates” feature. Any additional duplicates not recognized by the software were manually removed by the reviewers responsible for selecting articles for inclusion.

The selection of articles was carried out independently by the two reviewers (M.D. and C.D.R.). They initially listed the potentially eligible studies and then the included studies in two separate tables, which were subsequently compared. Potentially eligible studies were selected through title analysis, while the included studies were selected through full-text analysis and reading. Additionally, the inter-rater agreement between the two reviewers was assessed, and any disagreements were resolved by a third reviewer. Disagreements between reviewers in study selection were resolved through the following process: an independent re-evaluation of conflicting studies by each reviewer; discussions to deliberate on discrepancies; and, if consensus was not reached, a third reviewer was consulted to make the final decision.

### 2.4. Data Collection Process and Data Characteristics

The data to be reported in the tables from the included studies were determined during the preliminary drafting of the protocol. Similar to the records in the selection and screening phases, the data were independently extracted by the two reviewers and subsequently compared to reduce errors in data reporting. One of the two authors then consolidated the data into a single table or multiple tables, depending on the data type.

Discrepancies in data extraction between the two reviewers were initially identified and documented.

To resolve these discrepancies, the reviewers conducted a detailed discussion to compare the extracted data and clarify any misunderstandings or errors.

If, after discussion, no agreement was reached on certain data, the matter was referred to a third reviewer. The third reviewer reviewed the disputed entries and made the final decision on whether the data were extracted correctly. This process ensured that all discrepancies were resolved in a fair and transparent manner, based on the available evidence.

The data extracted from the articles included the first author, year of publication, study type, country conducting the study, number of patients, average age, gender, number of teeth with broken files, removal technique, presence of perforations, location of the separated fragment in the canal, number of failures or successes, and follow-up period.

### 2.5. Risk of Bias within Individual Studies and between Studies, Summary Measures, Summary of Results, Publication Bias, GRADE, and Trial Sequential Analysis

Particular attention was paid to the assessment of the risk of bias using a tool based on the checklist by McFarlane [12], and data from studies identified as high risk were excluded from the meta-analysis. As with other phases of the research, the risk of bias analysis was conducted independently by two reviewers, A.B. and M.D., followed by a comparison and consolidation of the findings into a single table.

Both the data and the results of the review were presented in tables. Aggregated data were represented not only numerically but also graphically through the creation of funnel plots, trial sequential analysis charts, and forest plots, which display the number of failures relative to the total number of retreated teeth and indices of heterogeneity such as Higgins’ I^2^.

Bias across studies was visually assessed by analyzing confidence interval (C.I.) overlaps and through the inconsistency index I^2^ (a value of I^2^ greater than 50% was considered moderate, and a random-effects analysis was applied in specific cases). If the meta-analysis showed high levels of heterogeneity, sensitivity analysis was planned by excluding studies with low C.I. overlap.

For the meta-analysis, particularly for the ratio of failed retreatments to retreated teeth, Open Meta-Analyst version 10 was utilized. Additionally, for the calculation of the pooled odds ratio, Review Manager 5.4 software was used (Cochrane Collaboration, Copenhagen, Denmark).

The odds ratio is a measure of the association between an exposure and an outcome. It represents the probability that an outcome will occur given a particular exposure, compared to the probabilities that the outcome will occur in the absence of that exposure. Instead, the aggregate odds ratio is a summary measure that combines the odds ratios from multiple studies, providing an overall estimate of the association between the exposure and the outcome. The pooled odds ratio is calculated using the Mantel–Haenszel method or the DerSimonian and Laird random-effects model (Open Meta-Analyst version 10, Review Manager 5.4).

The pooled odds ratio is typically presented in a forest plot, which visually displays the odds ratios of individual studies along with their confidence intervals and the overall pooled estimate. Each study is represented by a square (with dimensions proportional to its weight) and the aggregate estimate is represented by a diamond at the bottom of the graph.

The quality assessment of the evidence was conducted using online tools such as the GRADEpro Guideline Development Tool (GRADEpro-GDT, Evidence Prime, Web site: https://gdt.gradepro.org/, last access: 13 May 2024), and the results are reported in a table.

For the analysis of the power of the meta-analytic results, a free software tool for performing Trial Sequential Analysis (TSA) was used. This software, based on the Java programming language, is known as the “TSA software” (Copenhagen Trial Unit, Centre for Clinical Intervention Research, Copenhagen, Denmark, TSA software version: 0.9.5.10 Beta).

## 3. Results

### 3.1. Selection of Studies

The research question guiding the study selection was as follows: What is the rate of failure or success of endodontic retreatments in teeth with separated instruments within the roots?

The search phase involved consulting and extracting bibliographic references from two databases, SCOPUS (454 records) and PubMed (1155 records), and from the Cochrane library registry (11 trials), yielding a total of 1620 records. EndNote 8.0 software was used to identify duplicates using the “find duplicates” command and subsequently remove them, resulting in 1420 records. Any additional duplicates not identified by EndNote were manually removed by the reviewers during the article selection phase.

Specifically, the process of identifying and excluding duplicates was as follows:

Using reference management software (EndNote 8.0) to initially identify potential duplicates. This software compared titles, authors, and years of publication to find matches.

Manual verification by reviewers to confirm duplicates. Each potential duplicate identified by the software was carefully examined to ensure that it was indeed a duplicate of the same study.

Removal of confirmed duplicates before proceeding with the complete selection of texts. This removal was documented, and a log of exclusions was maintained for transparency.

After reviewing the title and abstract of each record, a total of 15 potentially eligible articles were identified, and at the end of the selection process, 11 articles were included for qualitative assessment, with 10 studies included in the meta-analysis.

Further exploration of the gray literature conducted on Google Scholar, Science Direct, and OpenGray using the keywords “broken separated endodontic” and “endodontic” in the DANS archive did not yield additional studies to include in the review (Figure 1). The records were screened by two authors (M.D. and C.D.R.) independently, and any uncertainties were addressed at the end of the selection process by involving a third author (A.B.) to resolve potential conflicts.

### 3.2. Data Characteristics

The articles included in the review are 11 and are as follows: Fu et al., 2011 [13], Maddalone and Gagliani, 2003 [14], Nevares, et al., 2012 [4], Malentacca et al., 2023 [15], Cuje et al., 2010 [16], Hülsmann and Schinkel, 1999 [17], Shen et al., 2004 [18], Terauchi et al., 2021 [19], Shiyakov and Vasileva, 2014 [20], Souter and Messer, 2005 [21], Suter et al., 2005, [3].

The extracted data are presented in two tables. Table 2 represents the data regarding the first author, the country of the study, the number of teeth with broken files, the technique adopted, the gender, the average age of patients, the number of failures, and the follow-up period. Meanwhile, Table 3 reports the data concerning the localization of fragments within the canal along with their respective numbers and the number of failures with any associated perforations. Only two studies did not report data on potential perforations: Terauchi et al., 2021 [19], and Shiyakov and Vasileva, 2014 [20].

The total number of teeth undergoing retreatment was approximately 1133 (1112 if the surgical approach is excluded); the number of failures was 172, while the number of perforations was 55, with 12 cases showing healing.

### 3.3. Risk of Bias

In Table 4, the assessment of risk of bias conducted by the two authors (M-D. and A.B.) is reported. The bias assessment tool was based on two checklists developed by Downs and Black [22] and Harvey [23], originally designed for observational and epidemiological studies including cross-sectional, cohort, and case-control studies. Adapted for use in conservative endodontic studies, similar to previous meta-analyses, these tools were utilized as alternatives to ROBINS or STROBE criteria.

In the table, numerical values ranging from 0 to 5 (where one = low and five = high) are reported for each parameter. The parameters assigned values from 0 to 5 are as follows:

(1) Non-response rate: Has the participation rate or follow-up rate been reported? Have the authors described efforts to increase the participation rate or follow-up? (5: Participation or follow-up rate is reported, and efforts to increase participation or follow-up are clearly described; 3–4: Participation or follow-up rate is reported, but efforts to increase participation or follow-up are only partially described; 1–2: Participation or follow-up rate is reported with minimal or no description of efforts to increase it; 0: Participation or follow-up rate is not reported, and no efforts to increase it are described).

(2) Representativeness of sample to target population: Were the subjects invited to participate in the study representative of the entire recruited population? (5: Subjects are fully representative of the target population; 3–4: Subjects are mostly representative, with minor deviations; 1–2: Subjects are somewhat representative, with significant deviations; 0: Subjects are not representative of the target population).

(3) Validity and reliability of outcome measurement: Were the primary outcome measures employed considered accurate (reliable and valid)? (5: Outcome measures are fully accurate (reliable and valid); 3–4: Outcome measures are mostly accurate, with minor concerns about reliability or validity; 1–2: Outcome measures have significant concerns about reliability or validity; 0: Outcome measures are neither reliable nor valid).

(4) Amount of loss to follow-up: Have the characteristics of non-participants/subjects lost to follow-up been described? Do the authors detail efforts made to increase the participation rate or follow-up? (5: Characteristics of non-participants/subjects lost to follow-up are fully described, and efforts to reduce loss to follow-up are detailed; 3–4: Characteristics of non-participants are partially described, with some efforts to reduce loss to follow-up detailed; 1–2: Minimal description of non-participants, with little effort to reduce loss to follow-up; 0: No description of non-participants and no efforts to reduce loss to follow-up).

(5) Appropriate statistical tests: Are the statistical methods adequately detailed? (5: Statistical methods are thoroughly detailed and appropriate; 3–4: Statistical methods are mostly detailed, with minor issues of appropriateness; 1–2: Statistical methods have significant issues in detail or appropriateness; 0: Statistical methods are inadequately detailed and inappropriate).

The risk of bias analysis was conducted by two authors. Each parameter for each study was assessed, and numerical values were assigned based on the criteria described above. The interpretation of the numerical values involved categorizing the overall risk of bias for each study as low (scores predominantly 4–5), moderate (scores predominantly 2–3), or high (scores predominantly 0–1). In cases where the authors initially disagreed on the risk of bias score, they discussed the discrepancies to reach a consensus. If consensus could not be achieved, a third reviewer was consulted to make the final decision.

The assessment was performed only for studies reporting data included in the meta-analysis (10 studies). Studies that exhibited clear issues during inclusion or data extraction were excluded from both the meta-analysis and bias assessment

### 3.4. Meta-Analysis

The meta-analysis of the data was carried out using two software tools: Open Meta-Analyst version 10, which generated forest plots to visualize the aggregated results, and Review Manager 5.4, developed by the Cochrane Collaboration in Copenhagen, Denmark. These tools were employed to synthesize and analyze the pooled data from the included studies, providing a comprehensive overview of the outcomes assessed.

The first meta-analysis conducted concerned the failure rate of retreatments in which there was a fragment of a separate instrument within the canal. There were 10 studies included in this meta-analysis: Fu et al., 2011 [13], Nevares, et al., 2012 [4], Malentacca et al., 2023 [15], Cuje et al., 2010 [16], Hülsmann and Schinkel, 1999 [17], Shen et al., 2004 [18], Terauchi et al., 2021 [19] Shiyakov and Vasileva, 2014 [20], Souter and Messer, 2005 [21], and Suter et al., 2005 [3].

The study by Maddalone and Gagliani, 2003 [14], was excluded from the meta-analysis due to the results being achieved through surgical endodontics, introducing a potential bias among the studies. Random effects were applied according to DerSimonian and Laird by calculating the aggregated ratio of the number of events (failures) over the total number of retreatments. The final ratio was 172/1012 (Figure 2).

The second meta-analysis conducted concerned the presence or occurrence of perforation in endodontically retreated teeth in which there was a fragment of a separate instrument inside the canal. There were eight studies included in this meta-analysis (Fu et al., 2011 [13], Nevares, et al., 2012 [4], Malentacca et al., 2023 [15], Cuje et al., 2010 [16], Hülsmann and Schinkel, 1999 [17], Shen et al., 2004 [18], Souter and Messer, 2005 [21], and Suter et al., 2005 [3]). Terauchi et al., 2021 [19] and Shiyakov and Vasileva, 2014 [20] were excluded because, in addition to not reporting perforations, they did not investigate their presence by reporting them in the reports. Random effects were applied according to DerSimonian and Laird by calculating the aggregate ratio between the number of perforations (failures) out of the total retreatment. The final ratio was 55/839 (Figure 3).

The third meta-analysis performed concerns the probability that a retreatment failure occurs when an instrument fragment is located in the middle and coronal third of the apical area. There were six studies included: Fu et al., 2011 [13], Malentacca et al., 2023 [15], Cuje et al., 2010 [16], Hülsmann and Schinkel, 1999 [17], Shen et al. 2004 [18], Souter and Messer, 2005 [21], and Suter et al., 2005 [3]. The ReV manager 5.4 program was used and fixed effects were applied. The OR (Odds Ratio) was 0.33 95% CI: [0.20, 0.53] clearly in favor of a (low) number of failures (Events) in the group in which the fragment was located coronally and in the middle third of the canal (Coronal/Middle) with a number of failures equal to 29 out of 330 compared to apical localization with 59 failures out of 278 retreatments. In fact, the central diamond that gives the size of the effect is clearly in favor of the Coronal/Middle group. The heterogeneity evaluated with an inconsistency index is low, with a Higgins index of I^2^ = 38% (Figure 4).

From the publication bias, assessed graphically through the creation of a funnel plot (Figure 5), the symmetry between the studies for the third meta-analysis can be inferred; however, the small number of included studies did not allow for an accurate assessment. Additionally, due to the low number of included studies, it was deemed unnecessary to perform a sensitivity analysis and subgroup analysis based on the techniques for removing the fractured instrument.

### 3.5. Trial Sequential Analysis, Grade

A sequential trial analysis (TSA) was conducted to assess the strength of the result of the third meta-analysis and adjust the findings to avoid type I and II errors. The program used was free Java software for TSA (TSA software version: 0.9.5.10 Beta). The O’Brien–Fleming spending function was applied using fixed effects.

The TSA results showed that despite only six studies being included, the data had statistical significance. Assuming a 50% Relative Risk Reduction (RRR) (a priori estimate) and 80% power (we used a maximum type I error of 5% and a maximum type II error of 20%), it was indeed highlighted how the blue line (Z-curve) crossed the sequential trial boundary (red sloping line) before reaching the optimal number of patients, which was determined to be 775 (Figure 6).

The authors also used GRADE pro-GDT to assess the quality of the evidence. The results suggest that the quality of evidence is moderate (Table 5).

## 4. Discussion

This study presents a systematic review with a meta-analysis to determine the success and failure rates of endodontically retreated teeth that contained a separated instrument prior to retreatment. Eleven studies were included in the review, with 10 studies contributing to the meta-analysis. The number of failures in instrument removal was 172 out of 1133 retreated teeth. Additionally, there were 55 perforations during retreatments, of which 12 resulted in favorable healing outcomes, representing approximately 30 percent of the perforated teeth.

In the literature, the proportion of instruments that fracture and become lodged within root canals during endodontic treatments is approximately 5% [1]. Moreover, it is evident that the teeth most commonly affected are lower molars, accounting for at least 50% of cases [5]. Anatomical factors appear to be crucial in the removal of separated instruments. Anatomical features such as straight canals and single-rooted teeth, as well as the location of the instrument before the curvature or in the coronal or middle third of the root canal, present significantly different success rates [17].

It is interesting to note that most failures in removing separated instruments occur when attempting to remove them from the apical third. The difficulty increases in the presence of curves when the separated instrument is no longer visible (beyond the root curvature) under the dental operating microscope.

Sauter and Messer reported that, out of a total of 60 cases, removal was successful in all cases where the separated endodontic instrument fragment was located in the coronal third (11 cases) or the middle third (22 cases), without perforations. In contrast, success was achieved in only 9 out of 27 cases involving the apical third, with an additional 7 resulting in perforations [21]. These findings are consistent with other studies included in this review [4].

The data from our meta-analysis clearly indicate that in retreatments, the odds ratio (OR) is 0.33 (Figure 4) for failures occurring when the separated instrument is located in the middle and coronal thirds compared to failures occurring in the apical third. Specifically, the number of failures in the middle and coronal thirds is 29 out of 330, while for the apical third, it is 59 out of 278 (in Figure 4, 60 events are reported as failures due to the addition of an event from the study by Cuje et al., 2010 [16], for correction when events are 0). The number of failures in attempting retreatment of the tooth and removal or bypassing of the instrument in the apical third is nearly double compared to the middle/coronal thirds. These data are further supported by the TSA, which confirmed the statistical power of the meta-analytical data by assuming a 50% Relative Risk Reduction (RRR) between instrument removal in the middle and coronal thirds versus removal in the apical third (Figure 6).

One challenge in removing the separated instrument within the canal is the excessive widening, leading to perforation and reduced tooth resistance, risking root fracture, especially when the fragment is in the apical or middle third. Moreover, the tooth’s prognosis significantly diminishes when the fragment remains, potentially leaving portions of the canal inadequately treated, leading to the wearing of radicular dentin and canal perforation.

Additionally, the meta-analysis data on perforation (Table 3, Figure 3) report 55 out of 839 incidents. However, root perforation does not necessarily result in failure (Table 3). Malentacca et al., 2023 [15] report 8 perforations, 2 in failed retreatments, while Fu et al., 2011 [13] mention 14 perforations with 8 successes and 6 failures.

Among the procedures commonly employed in the literature for removing fractured instruments is the technique extensively described by Ward et al. [24], a modification of that outlined by Ruddle [9] (Table 2).

This technique involves using fine ultrasonic tips, typically after creating a “staging platform” with a Gates Glidden bur (Dentsply Maillefer, Ballaigues, Switzerland) (Ward et al. recommend a size of 3 in the coronal portion and a size of 2 in the apical third as suggested by Cuje et al. [16]). This process is performed under an operating dental microscope to dislodge the instrument either by vibration or bypassing it with a small-sized endodontic file.

This technique proves effective in removing endodontic file fragments when partially located before the root canal curve in the straight portion, but success rates decrease when fragments are entirely within the curved segment, heightening the risk of root perforation. Cuje et al. [16], 2010, report that lower removal success rates are found in root canals with a curvature angle between 41° and 50° degrees.

Considerations should also be made regarding the alloy used to manufacture the endodontic file. Indeed, NiTi responds differently to ultrasonic vibration compared to steel. The direct application of ultrasonic vibration on a NiTi alloy may lead to instrument fracture at the point of contact with the ultrasonic tip [25].

The success of instrument removal must be confirmed with a periapical X-ray before canal filling. Success is achieved when the separated instrument is completely removed from the root canal without perforation, visually confirmed under a microscope or through canal bleeding or apex locator measurement. Other methods for fragment removal exist; for instance, Suter et al. (2005) describe at least four additional techniques apart from “Ultrasonics” in their clinical study [3].

Below are the main methodologies adopted in the included studies:✓The “Tube-and-Hedstrom file method” [26]: A circular groove is made around the separated instrument using ultrasonic tips. Then, a steel tube intercepts the instrument, followed by the insertion of a Hedstrom file into the tube, rotating it clockwise to trap the separated instrument between the steel tube and the Hedstrom file, and finally extracting all three components from the canal.✓Bypass: Generally performed with small-sized manual steel endodontic files, this technique is highly operator-dependent, relying on the tactile sensitivity and perseverance of the endodontist. It may lead to complications such as ledges, perforations, and transportation. This treatment choice can precede the use of ultrasonics in removal, allowing complete canal instrumentation and obturation with the fragment in situ. Incorporating the fragment into the root canal filling material significantly improves the case prognosis. This technique does not require direct visualization of the fragment and may be suitable when the fragment is located beyond a curve in the root canal [4].✓Forceps [26].✓The use of Masserann’s instrument [26].✓An ultrasonic device, operating microscope, and modified spinal needle [15].✓The loop technique [19].

The frequency of instrument fractures reported in the literature is further confirmed by Suter et al. (2005), who documented 97 cases of fractured instruments out of 1177 treated canals. Successful removal was achieved in 84 cases, while 7 cases resulted in root perforation and 6 in incomplete removal [3].

The meta-analysis data shown in Figure 2 indicate that the total number of failures in removing separated instruments was 172 out of 1012 retreated teeth, representing 17% of the retreatments. Notably, approximately two-thirds of these cases involved the separated instrument being located in the apical third (Figure 4).

Furthermore, the time taken to remove the separated instrument appears to be statistically associated with the success rate. According to Suter et al. (2005), the failure rate increases when the removal time exceeds 45–60 min. Additionally, Cuje et al. [16] and Hülsmann and Schinkel (1999) [17] provide noteworthy data regarding the length of the removed fragment. When the fragment length is less than 5 mm, the success rate is 97%, whereas for separated fragments with a length of 5–10 mm, the success rate drops to 76% [16].

Factors such as the type and root of the affected tooth, the length of the fractured instrument, the angle of curvature, and the removal time appear to be correlated with the success rate in retreatment. However, due to the limited number of studies and data included, it was not possible to perform a meta-analysis of these factors.

From the meta-analyses conducted in this review, significant findings include that the failure rate for retreating teeth with a fractured instrument in the canal is approximately 17%, with 172 failures out of 1012 retreatments. When the instrument is located in the apical third of the root, the failure rate increases to 21%, compared to only 8.8% in the middle and coronal thirds, with 29 failures out of 330 retreatments. Additionally, there were 55 perforations out of 839 retreatments, representing a perforation rate of 6.5%. Overall, the success–survivor rate for retreatment teeth with a fractured instrument is approximately 83%.

These findings can be valuable for clinicians, highlighting the significant variation in success rates depending on the fragment’s location. This information should be carefully considered when planning a retreatment.

The limitation of this systematic review lies in the small number of included studies and the heterogeneity of instruments and methods used, which prevents subgroup analysis. The use of a random-effects model helped mitigate this issue, yet high heterogeneity remained with an I^2^ of 95% (Figure 2) and I^2^ of 84% (Figure 3). To identify additional reports not published in major bibliometric databases (PubMed, Scopus), a search was also conducted in the grey literature using Google Scholar (which generally yields a vastly larger number of records, as shown in Figure 1, but is challenging to filter for relevant reports) and Open Grey (which, conversely, yields a much smaller number of records). The analysis of publication bias using the funnel plot (Figure 5) helps assess this limitation, but the small number of studies hampers a quantitative assessment of symmetry.

Moreover, although a thorough risk of bias assessment was conducted, the subjective nature of bias evaluation and potential inconsistencies in reporting standards across studies could influence the overall results. Additionally, the imaging techniques on which the accuracy of success and failure assessments is based can vary depending on the methods used (e.g., periapical radiography versus CBCT), which were not uniformly reported across all studies.

Periapical radiography, although widely used, has limitations in sensitivity and specificity, particularly in the detection of periapical lesions. Cone beam computed tomography (CBCT) has become a critical tool due to its increased accuracy in identifying and characterizing periapical pathology. CBCT provides three-dimensional images, allowing for a more precise evaluation of the root canal system and surrounding bony structures [27].

Questionable cases, such as those with ambiguous radiographic findings or conflicting clinical symptoms, require thorough evaluation using CBCT to ensure accurate diagnosis and treatment planning. Critical analysis of CBCT compared to periapical radiography shows that although CBCT is superior in terms of diagnostic accuracy, it should be used judiciously considering the higher radiation dose compared to traditional radiographs [28].

The integration of CBCT into endodontic practice improves the ability to evaluate the real extent of the pathology, thus improving the prognostic evaluation of endodontically treated teeth [28].

The findings of this study have several clinical implications: Highlighting the importance of the location of separated instruments within the root canal system, informing endodontists about the increased risk of failure associated with apically located separated instruments, and guiding treatment planning and patient counseling regarding the prognosis of endodontic retreatments with separated instruments.

## 5. Conclusions

In conclusion, the failure rate, which stands at approximately 17%, varies significantly depending on the location of the fractured instrument, ranging from 21% to 8.8%. Additionally, the incidence of perforations during the removal of the instrument is around 6.5%.

## Figures and Tables

**Figure 1 healthcare-12-01390-f001:**
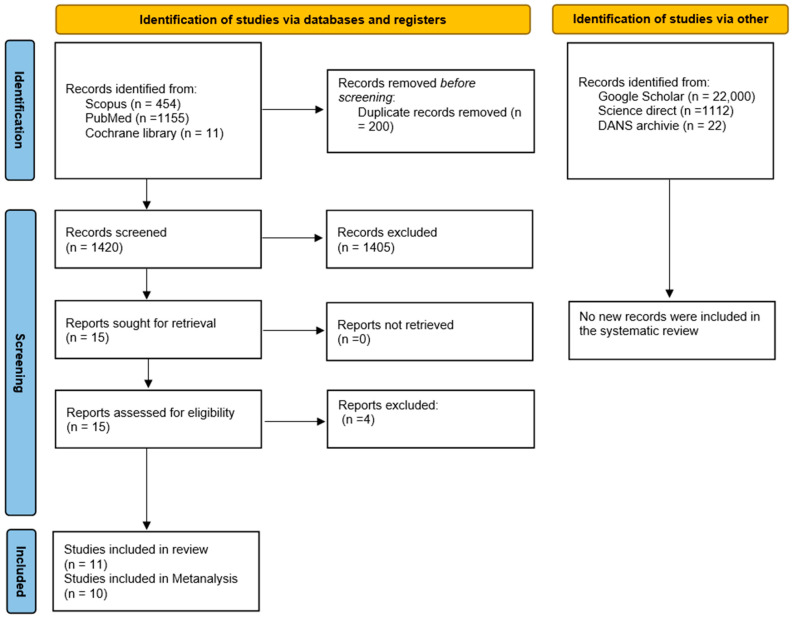
Flowchart of the article selection process.

**Figure 2 healthcare-12-01390-f002:**
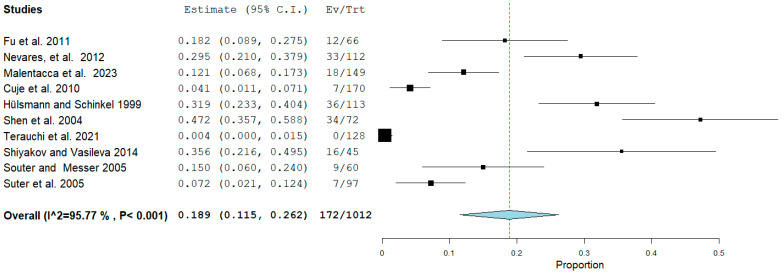
Binary random-effects model metric; proportion: 0.189; C.I. (Confidence Interval): (lower bound) 0.115 (upper bound) 0.262; *p*-value < 0.001; P = *p*-value; heterogeneity (Het.): tau^2: 0.012; df = degrees of freedom; Q = Q statistic; Q (df = 9), 212.870 Het. *p*-value: < 0.001, I^2^: 95.772; Standard error (SE): 0.037; I^2^ (I^2) = Higgins heterogeneity index, I^2^ < 50%, heterogeneity low. Each study graphically presents the first author’s name and publication date, alongside measurements such as the number of failures per total teeth and their relative proportions, complete with reported confidence intervals. The final summarized value, highlighted in bold, also includes its corresponding confidence intervals. A dashed red line denotes the average value position, while a light blue diamond represents the average effect measurement [3,4,13,15,16,17,18,19,20,21].

**Figure 3 healthcare-12-01390-f003:**
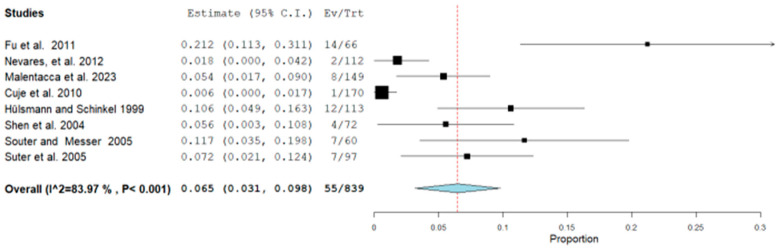
Binary random-effects model metric; proportion: 0.065; C.I. (Confidence Interval): (lower bound) 0.031 (upper bound) 0.098; *p*-value < 0.001; Standard error (SE): 0.017; I^2^ (I^2) = Higgins heterogeneity index, I^2^ < 50%, heterogeneity low; P = *p*-value; heterogeneity (Het.): tau^2: 0.002; Q = Q statistic; df = degrees of freedom; Q (df = 7), 43.664 Het. *p*-value: <0.001, I^2^: 83.969. Each graph of the study displays the first author’s name and publication date, along with measurements of the number of failures per total teeth and their relative proportions, including reported confidence intervals. The final value, along with its respective confidence intervals, is highlighted in bold. A dashed red line indicates the average value, while a light blue diamond represents the average effect measurement [3,4,13,15,16,17,18,21].

**Figure 4 healthcare-12-01390-f004:**
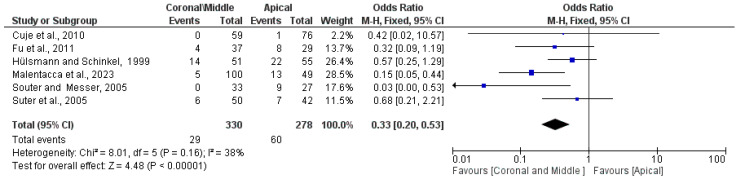
Forest plot of the fixed effects model of the meta-analysis; OR (Odds Ratio) = 0.33 95% CI: [0.20, 0.53]; df = degrees of freedom; I^2^ = Higgins heterogeneity index, I^2^ < 50%, heterogeneity irrelevant; I^2^ > 75%, significant heterogeneity; CI = confidence intervals; P = *p*-value; events = Failure. A correction factor of 1 was applied to studies with 0 events (failures) in the Apical group (Cuje et al., 2010 [16]). Each study graph displays the first author and publication date, along with the number of failures for each group, the total number of retreatments performed on teeth with separate instrument fragments, and the weight of each study expressed as a percentage. The final value, along with its relative confidence intervals, is presented in bold. A black line indicates the average value position, while a light black diamond represents the average effect measurement [3,13,15,16,17,21].

**Figure 5 healthcare-12-01390-f005:**
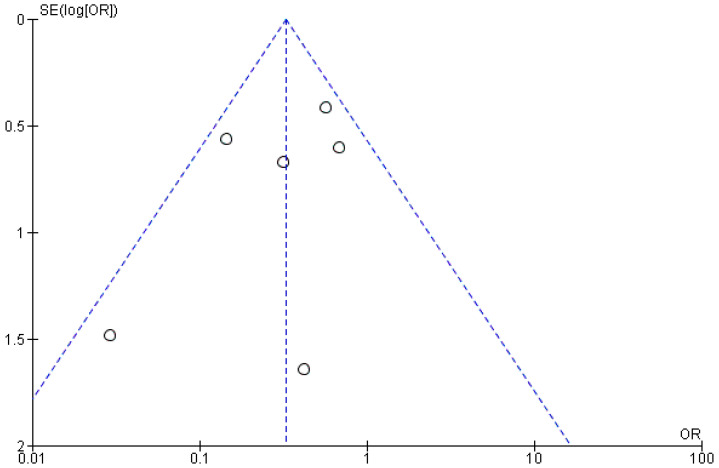
Funnel plot (RevManger 5.4): OR, odds ratio; SE, standard error. Graphically, there are no sources of heterogeneity. The presence of symmetry demonstrates the potential possibility of not presenting the bias of publication.

**Figure 6 healthcare-12-01390-f006:**
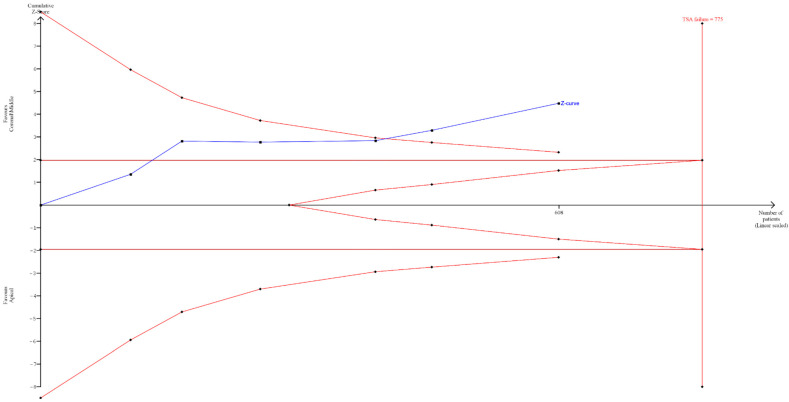
Sequential analysis of clinical studies where endodontic retreatment was performed with separated instruments located in the middle and coronal third of the tooth compared to the apical region. On the **left**, the inward-sloping red lines represent the boundaries of sequential trial monitoring. On the **right**, the outward-sloping red lines constitute the futility region. The continuous blue line is the cumulative Z-curve. Dark red line (Z = 1.98).

**Table 1 healthcare-12-01390-t001:** Search details.

	Search	Search Details
Pub Med	Sort by: Most Recent (separated instrument endodontic) OR (broken instrument endodontic) OR (Ultrasonic technique separated instrument)	((“divorce”[MeSH Terms] OR “divorce”[All Fields] OR “separated”[All Fields] OR “separation”[All Fields] OR “separations”[All Fields] OR “separabilities”[All Fields] OR “separability”[All Fields] OR “separable”[All Fields] OR “separate”[All Fields] OR “separately”[All Fields] OR “separates”[All Fields] OR “separating”[All Fields] OR “separational”[All Fields] OR “separative”[All Fields] OR “separator”[All Fields] OR “separators”[All Fields]) AND (“instrument”[All Fields] OR “instrument s”[All Fields] OR “instrumentation”[MeSH Subheading] OR “instrumentation”[All Fields] OR “instruments”[All Fields] OR “instrumented”[All Fields] OR “instrumenting”[All Fields]) AND (“endodontal”[All Fields] OR “endodontic”[All Fields] OR “endodontical”[All Fields] OR “endodontically”[All Fields] OR “endodontics”[MeSH Terms] OR “endodontics”[All Fields])) OR (“broken”[All Fields] AND (“instrument”[All Fields] OR “instrument s”[All Fields] OR “instrumentation”[MeSH Subheading] OR “instrumentation”[All Fields] OR “instruments”[All Fields] OR “instrumented”[All Fields] OR “instrumenting”[All Fields]) AND (“endodontal”[All Fields] OR “endodontic”[All Fields] OR “endodontical”[All Fields] OR “endodontically”[All Fields] OR “endodontics”[MeSH Terms] OR “endodontics”[All Fields])) OR ((“ultrasonically”[All Fields] OR “ultrasonicated”[All Fields] OR “ultrasonication”[All Fields] OR “ultrasonicator”[All Fields] OR “ultrasonics”[MeSH Terms] OR “ultrasonics”[All Fields] OR “ultrasonic”[All Fields]) AND (“methods”[MeSH Terms] OR “methods”[All Fields] OR “technique”[All Fields] OR “methods”[MeSH Subheading] OR “techniques”[All Fields] OR “technique s”[All Fields]) AND (“divorce”[MeSH Terms] OR “divorce”[All Fields] OR “separated”[All Fields] OR “separation”[All Fields] OR “separations”[All Fields] OR “separabilities”[All Fields] OR “separability”[All Fields] OR “separable”[All Fields] OR “separate”[All Fields] OR “separately”[All Fields] OR “separates”[All Fields] OR “separating”[All Fields] OR “separational”[All Fields] OR “separative”[All Fields] OR “separator”[All Fields] OR “separators”[All Fields]) AND (“instrument”[All Fields] OR “instrument s”[All Fields] OR “instrumentation”[MeSH Subheading] OR “instrumentation”[All Fields] OR “instruments”[All Fields] OR “instrumented”[All Fields] OR “instrumenting”[All Fields])) Translations separated: “divorce”[MeSH Terms] OR “divorce”[All Fields] OR “separated”[All Fields] OR “separation”[All Fields] OR “separations”[All Fields] OR “separabilities”[All Fields] OR “separability”[All Fields] OR “separable”[All Fields] OR “separate”[All Fields] OR “separately”[All Fields] OR “separates”[All Fields] OR “separating”[All Fields] OR “separational”[All Fields] OR “separative”[All Fields] OR “separator”[All Fields] OR “separators”[All Fields] instrument: “instrument”[All Fields] OR “instrument’s”[All Fields] OR “instrumentation”[Subheading] OR “instrumentation”[All Fields] OR “instruments”[All Fields] OR “instrumented”[All Fields] OR “instrumenting”[All Fields] endodontic: “endodontal”[All Fields] OR “endodontic”[All Fields] OR “endodontical”[All Fields] OR “endodontically”[All Fields] OR “endodontics”[MeSH Terms] OR “endodontics”[All Fields] instrument: “instrument”[All Fields] OR “instrument’s”[All Fields] OR “instrumentation”[Subheading] OR “instrumentation”[All Fields] OR “instruments”[All Fields] OR “instrumented”[All Fields] OR “instrumenting”[All Fields] endodontic: “endodontal”[All Fields] OR “endodontic”[All Fields] OR “endodontical”[All Fields] OR “endodontically”[All Fields] OR “endodontics”[MeSH Terms] OR “endodontics”[All Fields] Ultrasonic: “ultrasonically”[All Fields] OR “ultrasonicated”[All Fields] OR “ultrasonication”[All Fields] OR “ultrasonicator”[All Fields] OR “ultrasonics”[MeSH Terms] OR “ultrasonics”[All Fields] OR “ultrasonic”[All Fields] technique: “methods”[MeSH Terms] OR “methods”[All Fields] OR “technique”[All Fields] OR “methods”[Subheading] OR “techniques”[All Fields] OR “technique’s”[All Fields] separated: “divorce”[MeSH Terms] OR “divorce”[All Fields] OR “separated”[All Fields] OR “separation”[All Fields] OR “separations”[All Fields] OR “separabilities”[All Fields] OR “separability”[All Fields] OR “separable”[All Fields] OR “separate”[All Fields] OR “separately”[All Fields] OR “separates”[All Fields] OR “separating”[All Fields] OR “separational”[All Fields] OR “separative”[All Fields] OR “separator”[All Fields] OR “separators”[All Fields] instrument: “instrument”[All Fields] OR “instrument’s”[All Fields] OR “instrumentation”[Subheading] OR “instrumentation”[All Fields] OR “instruments”[All Fields] OR “instrumented”[All Fields] OR “instrumenting”[All Fields].
Scopus	Title, abstract, and keywords	TITLE-ABS-KEY ((broken OR separated) AND endodontic)
Cochrane Library	Title, abstract, and keywords	(broken OR separated) AND endodontic

**Table 2 healthcare-12-01390-t002:** Study characteristics and key data regarding the number of patients, the technique adopted, and results in terms of success or failure achieved. F (female), M (male), / (Data not reported), m (months), y (years).

First Authors, Data, Reference	Nationality	Type of Study	Number of Teeth with Broken Files	Technique	Gender, Medium Age	Outcome	Follow Up
Fu et al., 2011 [13]	China	Retrospective	66	Ultrasonic technique	F = 53,/ M = 13,/	56 removed 8 failed	12–68 m
Maddalone & Gagliani [14] 2003	Italy	Prospective	120	Apical surgery	F = 58, 36 M = 28, 45	94 healed 17 healed with scar 4 uncertain 5 failed	3 y
Nevares, et al., 2012 [4]	Brasile	Prospective	112	Removal or bypass	/	37 removed 42 bypassed 33 failed	/
Malentacca et al., 2023 [15]	Italy	Retrospective	158	Needle technique or bypass	^1^ F = 96,/ M = 53,/	131 removed 9 bypassed 18 failed	1–5 y
Cuje et al., 2010 [16]	Germany	Clinical study	170	Ultrasonic	/	163 healed 7 failed	/
Hülsmann and Schinkel, 1999 [17]	Germany	Comparative in-vivo and ex-vivo study	105 teeth, 113 fragments	Canal finder system and ultrasonic	/	77 removed 22 bypassed 36 failed	/
Shen et al., 2004 [18]	China	Clinical study	72	Removing or bypassing	/	32 removed 8 bypassed 34 failed	/
Terauchi et al., 2021 [19]	Japan	Prospective	128	Ultrasonic and loop device	F: 102 M: 26 Medium age: 47.7	0 failed 89.8% success using only the ultrasonic tip	6 m
Shiyakov and Vasileva, 2014 [20]	Bulgaria	Retrospective	45	Ultrasonic or Bypass	/	16 failed 22 removed 7 bypassed	/
Souter and Messer, 2005 [21]	Australia	Vitro, case series	60	Ultrasonic	/	42 removed 18 failed	/
Suter et al., 2005 [3]	Switzerland	Retrospective	97	Ultrasonic	/	84 removed 13 failed	/

^1^ Gender data do not refer to patients who were bypassed by the separate tool.

**Table 3 healthcare-12-01390-t003:** Localization of the fractured instruments within the root canal by success rate.

First Authors, Data	Teeth	Third, Location of Fragment	Teeth	Not Removed, Not Bypass or Failed	Perforation
Fu et al., 2011 [13]	66	Coronal, Middle	37	4	14 (8 healed)
		Apical	29	8	
Nevares, et al., 2012 [4]	112	\	112	33	2 (1 apical)
Malentacca et al., 2023 [15]	149	Coronal	13	1	8 (2 failed)
		Middle	87	4	
		Apical	49	13	
Cuje et al., 2010 ^1^ [16]	170	Coronal	15	0	
		Middle	44	0	
		Apical	54	3	
		Beyond foram, apical foram	1	1	1
		Whole length of root canal	7	0	
		Coronal and Middle third	17	0	
		Middle and apical third	32	3	
Hülsmann and Schinkel, 1999 [17]	113	Coronal	5	0	12
		Middle	44	14	
		Apical	44	18	
		Beyond foram, apical foram	7	4	
		Whole length of root canal	7	0	
		Coronal and Middle third	2	0	
		Middle and apical third	4	0	
Shen et al., 2004 [18]	72	Before curvature	6	0	4
		At the curvature	30	12	
		Beyond curvature	32	22	
		Straight canals	4	0	
Terauchi et al., 2021 [19]	128	\	\	0	\
Shiyakov and Vasileva, 2014 [20]	45	\	\	16	\
Souter and Messer, 2005 [21]	60	Coronal	11	0	0
		Middle	22	0	0
		Apical	27	9	7
Suter et al., 2005 [3]	97	Coronal	19	2	7 (4 Apical)
		Middle	31	4	
		Apical	40	5	
		Apical foram	2	2	
		Whole length of root canal	5	0	

^1^ Between failures, the authors report a bypass of the instrument.

**Table 4 healthcare-12-01390-t004:** Assessment of risk of bias within the studies.

	Selection		Outcome	Loss to Follow-Up	Analysis	Score
Reference	Non Response Rate	Representativeness of Sample to Target Population	Validity and Reliability of Outcome Measurement	Amount of Loss to Follow-Up	Appropriate Statistical Tests	
Fu et al., 2011 [13]	5	5	5	5	4	24
Nevares, et al., 2012 [4]	3	4	5	2	4	18
Malentacca et al., 2023 [15]	5	5	5	5	4	24
Cuje et al., [16] 2010,	3	4	5	4	4	20
Hülsmann and Schinkel, 1999 [17]	3	5	4	2	4	18
Shen et al., 2004 [18]	4	4	4	4	4	20
Terauchi et al., 2021 [19]	5	4	3	5	4	21
Shiyakov and Vasileva, 2014 [20]	3	4	3	3	3	16
Souter and Messer, 2005 [21]	1	5	5	2	3	17
Suter et al., 2005 [3]	4	5	5	3	5	22

**Table 5 healthcare-12-01390-t005:** Evaluation of GRADE pro GDT: ⊕◯◯◯ Very low, ⊕⊕◯◯ Low, ⊕⊕⊕◯ Moderate, ⊕⊕⊕⊕ High, CI (confidence Interval).

Certainty Assessment							Summary of Findings				
Participants (studies) Follow-up	Risk of bias	Inconsistency	Indirectness	Imprecision	Publication bias	Overall certainty of evidence	Study event rates (%)		Relative effect (95% CI)	Anticipated absolute effects	
							With Apical	With Coronal\Middle		Risk with Apical	Risk difference with Coronal\Middle
608 (6 non-randomised studies)	not serious	not serious	not serious	not serious	strong association	⨁⨁⨁◯ Moderate	60/278 (21.6%)	29/330 (8.8%)	OR 0.33 (0.20 to 0.53)	60/278 (21.6%)	133 fewer per 1000 (from 164 fewer to 89 fewer)

## Data Availability

Not applicable.

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
