# Peer review of "Analysis of Endodontic Successes and Failures in the Removal of Fractured Endodontic Instruments during Retreatment: A Systematic Review, Meta-Analysis, and Trial Sequential Analysis"

_healthcare, 2024, doi:10.3390/healthcare12141390_

Round 1
Reviewer 1 Report
Comments and Suggestions for Authors
Dear Authors,
I have reviewed your manuscript titled “Analysis of Endodontic Successes and Failures in the Removal of Fractured Endodontic Instruments During Retreatment: A Systematic Review, Meta-analysis and Trial Sequential Analys” with great interest. I appreciate the effort put into this study and the potential implications of your findings.
Overall:
The research question is the success rate of the removal of fractured endodontic instruments during retreatment. The focus is the impact of the location within the root (apical, middle, coronal) for a successful retreatment of the fragment. The main result is the logical fact that it is harder to remove a fractured fragment at the apical part.
More in detail:
1.Introduction
- the introduction would benefit from a more detailed explanation of the leading selection criteria for the included reviews
2. Materials and methods
- search terms should be included in a structured table and not in the text
- the explanation of the search terms should be more precise
- please provide more detailed criteria for excluding studies, especially for the high dropout rate here
- try do describe the process for resolving disagreements between reviewers in more detail
- the data extraction process should be more detailed and transparent, e.g. how discrepancies were managed
3. Results
- the process of identifying and excluding duplicates needs to be described in detail, how was the removement conducted?
- criteria for assessing the risk of bias should be explained in detail, how were the numerical values assigned and interpreted?
4. Discussion
- the statistical methods should be more described, especially the calculation of odds ratios and relative risk reductions
- include potential limitations and success rates
- expand the discussion on clinical implications of the findings
Author Response
Dear Authors,
I have reviewed your manuscript titled “Analysis of Endodontic Successes and Failures in the Removal of Fractured Endodontic Instruments During Retreatment: A Systematic Review, Meta-analysis and Trial Sequential Analys” with great interest. I appreciate the effort put into this study and the potential implications of your findings.
Overall:
The research question is the success rate of the removal of fractured endodontic instruments during retreatment. The focus is the impact of the location within the root (apical, middle, coronal) for a successful retreatment of the fragment. The main result is the logical fact that it is harder to remove a fractured fragment at the apical part.
ANSWER
Dear Reviewer,
Thank you very much for your thorough review of our manuscript titled “Analysis of Endodontic Successes and Failures in the Removal of Fractured Endodontic Instruments During Retreatment: A Systematic Review, Meta-analysis and Trial Sequential Analysis.” We greatly appreciate your detailed feedback and the constructive suggestions provided.
Your insights and comments have been invaluable in refining our manuscript, and we are grateful for the time and effort you have invested in reviewing our work. We have addressed the points you raised and believe these changes have significantly improved the clarity and quality of our study.
Thank you once again for your thoughtful review and for your support in enhancing the scientific value of our research.
More in detail:
1.Introduction
- the introduction would benefit from a more detailed explanation of the leading selection criteria for the included reviews
ANSWER
Further explanation of the inclusion criteria has been added to the introduction as requested. Below is the part added to the manuscript.
The reviewers established primary criteria for the selection and inclusion of studies to address the objectives of the systematic review, based on several factors. Studies were included and considered in the review if they: evaluated the outcomes of endodontic retreatment where separated instruments were present; provided sufficient data on success and failure rates, including the specific locations of separated instruments within the root canals; used consistent and comparable criteria for defining the success and failure of endodontic treatment; and were published in peer-reviewed medical scientific journals and available in English. These criteria were chosen to ensure that the meta-analysis synthesised relevant and high-quality data, allowing for a thorough analysis of the impact of separated instruments on the success rates of retreatment
- Materials and methods
- search terms should be included in a structured table and not in the text
ANSWER
Search details have been removed from the text and are provided in a table as required. Below is the part added to the manuscript.
|
search |
Search details |
Pub Med |
Sort by: Most Recent (separated instrument endodontic) OR (broken instrument endodontic) OR (Ultrasonic technique separated instrument) |
(("divorce"[MeSH Terms] OR "divorce"[All Fields] OR "separated"[All Fields] OR "separation"[All Fields] OR "separations"[All Fields] OR "separabilities"[All Fields] OR "separability"[All Fields] OR "separable"[All Fields] OR "separate"[All Fields] OR "separately"[All Fields] OR "separates"[All Fields] OR "separating"[All Fields] OR "separational"[All Fields] OR "separative"[All Fields] OR "separator"[All Fields] OR "separators"[All Fields]) AND ("instrument"[All Fields] OR "instrument s"[All Fields] OR "instrumentation"[MeSH Subheading] OR "instrumentation"[All Fields] OR "instruments"[All Fields] OR "instrumented"[All Fields] OR "instrumenting"[All Fields]) AND ("endodontal"[All Fields] OR "endodontic"[All Fields] OR "endodontical"[All Fields] OR "endodontically"[All Fields] OR "endodontics"[MeSH Terms] OR "endodontics"[All Fields])) OR ("broken"[All Fields] AND ("instrument"[All Fields] OR "instrument s"[All Fields] OR "instrumentation"[MeSH Subheading] OR "instrumentation"[All Fields] OR "instruments"[All Fields] OR "instrumented"[All Fields] OR "instrumenting"[All Fields]) AND ("endodontal"[All Fields] OR "endodontic"[All Fields] OR "endodontical"[All Fields] OR "endodontically"[All Fields] OR "endodontics"[MeSH Terms] OR "endodontics"[All Fields])) OR (("ultrasonically"[All Fields] OR "ultrasonicated"[All Fields] OR "ultrasonication"[All Fields] OR "ultrasonicator"[All Fields] OR "ultrasonics"[MeSH Terms] OR "ultrasonics"[All Fields] OR "ultrasonic"[All Fields]) AND ("methods"[MeSH Terms] OR "methods"[All Fields] OR "technique"[All Fields] OR "methods"[MeSH Subheading] OR "techniques"[All Fields] OR "technique s"[All Fields]) AND ("divorce"[MeSH Terms] OR "divorce"[All Fields] OR "separated"[All Fields] OR "separation"[All Fields] OR "separations"[All Fields] OR "separabilities"[All Fields] OR "separability"[All Fields] OR "separable"[All Fields] OR "separate"[All Fields] OR "separately"[All Fields] OR "separates"[All Fields] OR "separating"[All Fields] OR "separational"[All Fields] OR "separative"[All Fields] OR "separator"[All Fields] OR "separators"[All Fields]) AND ("instrument"[All Fields] OR "instrument s"[All Fields] OR "instrumentation"[MeSH Subheading] OR "instrumentation"[All Fields] OR "instruments"[All Fields] OR "instrumented"[All Fields] OR "instrumenting"[All Fields])) Translations separated: "divorce"[MeSH Terms] OR "divorce"[All Fields] OR "separated"[All Fields] OR "separation"[All Fields] OR "separations"[All Fields] OR "separabilities"[All Fields] OR "separability"[All Fields] OR "separable"[All Fields] OR "separate"[All Fields] OR "separately"[All Fields] OR "separates"[All Fields] OR "separating"[All Fields] OR "separational"[All Fields] OR "separative"[All Fields] OR "separator"[All Fields] OR "separators"[All Fields] instrument: "instrument"[All Fields] OR "instrument's"[All Fields] OR "instrumentation"[Subheading] OR "instrumentation"[All Fields] OR "instruments"[All Fields] OR "instrumented"[All Fields] OR "instrumenting"[All Fields] endodontic: "endodontal"[All Fields] OR "endodontic"[All Fields] OR "endodontical"[All Fields] OR "endodontically"[All Fields] OR "endodontics"[MeSH Terms] OR "endodontics"[All Fields] instrument: "instrument"[All Fields] OR "instrument's"[All Fields] OR "instrumentation"[Subheading] OR "instrumentation"[All Fields] OR "instruments"[All Fields] OR "instrumented"[All Fields] OR "instrumenting"[All Fields] endodontic: "endodontal"[All Fields] OR "endodontic"[All Fields] OR "endodontical"[All Fields] OR "endodontically"[All Fields] OR "endodontics"[MeSH Terms] OR "endodontics"[All Fields] Ultrasonic: "ultrasonically"[All Fields] OR "ultrasonicated"[All Fields] OR "ultrasonication"[All Fields] OR "ultrasonicator"[All Fields] OR "ultrasonics"[MeSH Terms] OR "ultrasonics"[All Fields] OR "ultrasonic"[All Fields] technique: "methods"[MeSH Terms] OR "methods"[All Fields] OR "technique"[All Fields] OR "methods"[Subheading] OR "techniques"[All Fields] OR "technique's"[All Fields] separated: "divorce"[MeSH Terms] OR "divorce"[All Fields] OR "separated"[All Fields] OR "separation"[All Fields] OR "separations"[All Fields] OR "separabilities"[All Fields] OR "separability"[All Fields] OR "separable"[All Fields] OR "separate"[All Fields] OR "separately"[All Fields] OR "separates"[All Fields] OR "separating"[All Fields] OR "separational"[All Fields] OR "separative"[All Fields] OR "separator"[All Fields] OR "separators"[All Fields] instrument: "instrument"[All Fields] OR "instrument's"[All Fields] OR "instrumentation"[Subheading] OR "instrumentation"[All Fields] OR "instruments"[All Fields] OR "instrumented"[All Fields] OR "instrumenting"[All Fields].
|
Scopus |
title, abstract, and keywords |
TITLE-ABS-KEY ( ( broken OR separated ) AND endodontic ) |
Cochrane Library |
title, abstract, and keywords |
(broken OR separated) AND endodontic |
- the explanation of the search terms should be more precise
ANSWER
A more precise explanation in the choice of search terms as requested has been provided. Below is the part added to the manuscript.
The search terms were chosen to encompass the various terminologies used in the literature for endodontic retreatment and the presence of separated instruments. The terms were combined using Boolean operators to ensure comprehensive coverage of relevant studies.
- please provide more detailed criteria for excluding studies, especially for the high dropout rate here
ANSWER
More details regarding the exclusion criteria and in particular the high dropout rate were provided as requested. Below is the part added to the manuscript.
Studies were excluded from the review if they had any of the following criteria: High dropout rate: studies in which more than 20% of initial participants dropped out of the study before its conclusion. This criterion was chosen because a high dropout rate may compromise the validity and reliability of the study results. Insufficient data: Studies that did not provide sufficient data on the success and failure rates of endodontic retreatment, including details on the location of individual instruments within root canals. Inconsistent definitions of success and failure: Studies that used non-standardized or non-comparable criteria to define success and failure of endodontic treatment.
Furthermore, the following exclusion criteria were applied: excluding all reports related to systematic reviews, scoping reviews, mapping reviews, narrative reviews, case reports, case series with a low number of included cases, in vitro and in silico studies, studies not reporting data on the failures or successes of retreatment, or where it was not possible to extract data regarding teeth containing separated instruments or their eventual removal, studies published in a language other than English, and those lacking an abstract in English, deemed to be at high risk of bias.
- try do describe the process for resolving disagreements between reviewers in more detail
ANSWER
The description of the process for resolving disagreements between reviewers has been added in more detail as requested. Below is the part added to the manuscript.
The selection of articles was carried out independently by the two reviewers (M.D. and C.D.R.). They initially listed the potentially eligible studies and then the included studies in two separate tables, which were subsequently compared. Potentially eligible studies were selected through title analysis, while included studies were selected through full-text analysis and reading. Additionally, the inter-rater agreement between the two reviewers was assessed, and any disagreements were resolved by a third reviewer. Disagreements between reviewers in study selection were resolved through the following process: Independent re-evaluation of conflicting studies by each reviewer; Discussions to deliberate on discrepancies; If consensus was not reached, a third reviewer was consulted to make the final decision.
- the data extraction process should be more detailed and transparent, e.g. how discrepancies were managed
ANSWER
the data extraction process was reported in a more detailed and transparent way, especially how discrepancies were handled. Below is the part added to the manuscript
The data to be reported in the tables from the included studies were determined during the preliminary drafting of the protocol. Similar to the records in the selection and screening phases, the data were independently extracted by the two reviewers and subsequently compared to reduce errors in data reporting. One of the two authors then consolidated the data into a single table or multiple tables, depending on the data type.
Discrepancies in data extraction between the two reviewers were initially identified and documented.
To resolve these discrepancies, the reviewers conducted a detailed discussion to compare the extracted data and clarify any misunderstandings or errors.
If, after discussion, no agreement was reached on certain data, the matter was referred to a third reviewer. The third reviewer reviewed the disputed entries and made the final decision on whether the data was extracted correctly. This process ensured that all discrepancies were resolved in a fair and transparent manner, based on available evidence.
The extracted data from the articles included the first author, year of publication, study type, country conducting the study, number of patients, average age, gender, number of teeth with broken files, removal technique, presence of perforations, location of the separated fragment in the canal, number of failures or successes, and follow-up period.
- Results
- the process of identifying and excluding duplicates needs to be described in detail, how was the removement conducted?
ANSWER
the process of identifying and excluding duplicates and how the removal was carried out was described in detail as required. Below is the part added to the manuscript
The research question guiding the study selection was as follows: What is the rate of failures or successes of endodontic retreatments in teeth with separated instruments within the roots?
The search phase involved consulting and extracting bibliographic references from two databases, SCOPUS (454 records) and PubMed (1155 records), and from the Cochrane library registry (11 trials), yielding a total of 1620 records. The EndNote 8.0 software was used to identify duplicates using the "find duplicates" command and subsequently remove them, resulting in 1420 records. Any additional duplicates not identified by EndNote were manually removed by the reviewers during the article selection phase.
Specifically, the process of identifying and excluding duplicates involved:
Using reference management software (EndNote 8.0) to initially identify potential duplicates. This software compared titles, authors, and years of publication to find matches.
Manual verification by reviewers to confirm duplicates. Each potential duplicate identified by the software was carefully examined to ensure that it was indeed a duplicate of the same study.
Removal of confirmed duplicates before proceeding with the complete selection of texts. This removal was documented, and a log of exclusions was maintained for transparency.
After reviewing the title and abstract of each record, a total of 15 potentially eligible articles were identified, and at the end of the selection process, 11 articles were included for qualitative assessment, with 10 studies included in the meta-analysis.
- criteria for assessing the risk of bias should be explained in detail, how were the numerical values assigned and interpreted?
ANSWER
a detailed explanation was provided as required for the criteria for assessing risk of bias and in detail how the numerical values were assigned and interpreted. Below is the part added to the manuscript.
In the table, numerical values ranging from 0 to 5 (where one = low and five = high) are reported for each parameter. The parameters assigned values from 0 to 5 are as follows:
(1) Non-response rate: Has the participation rate or follow-up rate been reported? Have the authors described efforts to increase the participation rate or follow-up? (5: Participation or follow-up rate is reported, and efforts to increase participation or follow-up are clearly described. 3-4: Participation or follow-up rate is reported, but efforts to increase participation or follow-up are partially described. 1-2: Participation or follow-up rate is reported with minimal or no description of efforts to increase it. 0: Participation or follow-up rate is not reported, and no efforts to increase it are described).
(2) Representativeness of sample to target population: Were the subjects invited to participate in the study representative of the entire recruited population? (5: Subjects are fully representative of the target population. 3-4: Subjects are mostly representative, with minor deviations. 1-2: Subjects are somewhat representative, with significant deviations. 0: Subjects are not representative of the target population)
(3) Validity and reliability of outcome measurement: Were the primary outcome measures employed considered accurate (reliable and valid)? (5: Outcome measures are fully accurate (reliable and valid).3-4: Outcome measures are mostly accurate, with minor concerns about reliability or validity.1-2: Outcome measures have significant concerns about reliability or validity.0: Outcome measures are neither reliable nor valid.)
(4) Amount of loss to follow-up: Have the characteristics of non-participants/subjects lost to follow-up been described? Do the authors detail efforts made to increase the participation rate or follow-up? (5: Characteristics of non-participants/subjects lost to follow-up are fully described, and efforts to reduce loss to follow-up are detailed. 3-4: Characteristics of non-participants are partially described, with some efforts to reduce loss to follow-up detailed. 1-2: Minimal description of non-participants, with little effort to reduce loss to follow-up. 0: No description of non-participants and no efforts to reduce loss to follow-up.)
(5) Appropriate statistical tests: Are the statistical methods adequately detailed? (5: Statistical methods are thoroughly detailed and appropriate. 3-4: Statistical methods are mostly detailed, with minor issues in appropriateness. 1-2: Statistical methods have significant issues in detail or appropriateness. 0: Statistical methods are inadequately detailed and inappropriate.)
The risk of bias analysis was conducted by two authors. Each parameter for each study was assessed, and numerical values were assigned based on the criteria described above. The interpretation of the numerical values involved categorising the overall risk of bias for each study as low (scores predominantly 4-5), moderate (scores predominantly 2-3), or high (scores predominantly 0-1). In cases where the authors initially disagreed on the risk of bias score, they discussed the discrepancies to reach a consensus. If consensus could not be achieved, a third reviewer was consulted to make the final decision.
Assessment was performed only for studies reporting data included in the meta-analysis (10 studies). Studies that exhibited clear issues during inclusion or data extraction were excluded from both the meta-analysis and bias assessment
- Discussion
- the statistical methods should be more described, especially the calculation of odds ratios and relative risk reductions.
ANSWER
the statistical methods have been further described providing more details on the pooled calculation, the odds ratio for the relative risk reductions has not been included as it has not been the subject of analysis. Below is the part added to the manuscript.
The odds ratio is a measure of association between an exposure and an outcome. It represents the probabilities that an outcome will occur given a particular exposure, compared to the probabilities that the outcome will occur in the absence of that exposure. Instead, the aggregate odds ratio is a summary measure that combines the odds ratios from multiple studies, providing an overall estimate of the association between exposure and outcome. The pooled odds ratio is calculated using the Mantel-Haenszel method or the DerSimonian and Laird random effects model.
The pooled odds ratio is typically presented in a forest plot, which visually displays the odds ratios of individual studies along with their confidence intervals and the overall pooled estimate. Each study is represented by a square (with dimensions proportional to its weight) and the aggregate estimate is represented by a diamond at the bottom of the graph
- include potential limitations and success rates
ANSWER
The limitations of the systematic review have been expanded as requested. Below is the part added to the manuscript. Overall, the success rate for retreating teeth with a fractured instrument is approximately 83% .Below is the part added to the manuscript.
Additionally, there were 55 perforations out of 839 retreatments, representing a perforation rate of 6.5%. Overall, the success rate for retreatment teeth with a fractured instrument is approximately 83%.
The limitation of this systematic review lies in the small number of included studies and the heterogeneity of instruments and methods used, which prevents subgroup analysis. The use of a random-effects model helped mitigate this issue, yet high heterogeneity remained with an I2 of 95% (Figure 2) and I2 of 84% (Figure 3).To identify additional reports not published in major bibliometric databases (PubMed, Scopus), a search was also conducted in the grey literature using Google Scholar (which generally yields a vastly larger number of records, Figure 1, but is challenging to filter for relevant reports) and Open Grey (which, conversely, yields a much smaller number of records). The analysis of publication bias using the funnel plot (Figure 5) helps assess this limitation, but the small number of studies hampers a quantitative assessment of symmetry.
Moreover, although a thorough risk of bias assessment was conducted, the subjective nature of bias evaluation and potential inconsistencies in reporting standards across studies could influence the overall results. Additionally, the imaging techniques on which the accuracy of success and failure assessments is based can vary depending on the methods used (e.g., periapical radiography versus CBCT), which were not uniformly reported across all studies.
- expand the discussion on clinical implications of the findings
ANSWER
the discussion was expanded on the clinical implications of the findings as requested. Below is the part added to the manuscript.
The findings of this study have several clinical implications: Highlighting the importance of the location of separated instruments within the root canal system; Informing endodontist about the increased risk of failure associated with apically located separated instruments; Guiding treatment planning and patient counseling regarding the prognosis of endodontic retreatments with separated instruments.

Reviewer 2 Report
Comments and Suggestions for Authors
This study presents a systematic review with meta-analysis to evaluate the success rates of endodontic retreatments in teeth where separated instruments are located within the roots. The meta-analysis results indicate that failures are more frequent when instruments are located in the apical third, with a failure rate of 21%, compared to an 8.8% failure rate in the middle/coronal third. The anatomy of the root canals, particularly the location of the separated instruments, significantly influences the success rates.
Comments:
1. The introduction to the study provides a contextualization of the topic; however, this topic must clearly address the gap in the literature and provide a strong justification.
2. The search for studies and statistical treatment and results have no comments.
3. Authors must avoid using the first person plural (We) throughout the text. The first sentence of the discussion can be removed (We conducted a systematic review with a meta-analysis to determine the success and failure rates of endodontically retreated teeth that contained a separated instrument prior to retreatment).
4. In this type of study, discussing endodontic success and failure, it is essential to characterize the criteria that are currently adopted regarding success and failure, in addition to addressing questionable cases and establishing a critical analysis of the accuracy between periapical radiography and CBCT. These criteria must be discussed within a current context, involving imaging examination methods with cone beam computed tomography.
5. Make sure all the text follows the style of the journal. Correct references 7, 12, and 20.
The study is interesting and contributes to the field, but implementing these observations can make it more appealing to readers.
Author Response
This study presents a systematic review with meta-analysis to evaluate the success rates of endodontic retreatments in teeth where separated instruments are located within the roots. The meta-analysis results indicate that failures are more frequent when instruments are located in the apical third, with a failure rate of 21%, compared to an 8.8% failure rate in the middle/coronal third. The anatomy of the root canals, particularly the location of the separated instruments, significantly influences the success rates.
ANSWER
Dear Reviewer,
Thank you very much for your thorough review of our manuscript titled “Analysis of Endodontic Successes and Failures in the Removal of Fractured Endodontic Instruments During Retreatment: A Systematic Review, Meta-analysis and Trial Sequential Analysis.” We greatly appreciate your detailed feedback and the constructive suggestions provided.
Your insights and comments have been invaluable in refining our manuscript, and we are grateful for the time and effort you have invested in reviewing our work. We have addressed the points you raised and believe these changes have significantly improved the clarity and quality of our study.
Thank you once again for your thoughtful review and for your support in enhancing the scientific value of our research.
Comments:
- The introduction to the study provides a contextualization of the topic; however, this topic must clearly address the gap in the literature and provide a strong justification.
ANSWER
A clarification on the gap in the literature has been provided as requested Below is the part added to the manuscript
Conversely, McGuigan et al. focused on the incidence of instrument separation and the types of instruments, reporting that the incidence of instrument separation in NiTi instruments is similar to that in stainless steel (SS) files (1).
Despite the technological advancements in endodontics and materials, the presence of separated instruments within the canals remains a primary challenge for practitioners involved in endodontic retreatments. The current medical scientific literature offers various guidelines on the management and outcomes of endodontic retreatment; however, there is a notable gap concerning the specific success and failure rates associated with teeth containing separated instruments. While some studies have addressed the general outcomes of retreatment, comprehensive analyses focusing on the impact of separated instruments are limited.
This gap in the literature underscores the need for a systematic review and meta-analysis to aggregate existing data and provide a clearer understanding of the prognostic implications of separated instruments in endodontic retreatment. By synthesising and aggregating the findings of multiple studies, this research aims to offer valuable insights into the factors influencing success and failure rates, thereby guiding clinical decision-making and improving patient outcomes.
This study aims to conduct a systematic literature review to evaluate the failure rate of endodontic retreatments that involved a separated instrument within the root canal prior to retreatment. The reviewers established primary criteria for the selection and inclusion of studies to address the objectives of the systematic review, based on several factors. Studies were included and considered in the review if they: evaluated the outcomes of endodontic retreatment where separated instruments were present; provided sufficient data on success and failure rates, including the specific locations of separated instruments within the root canals; used consistent and comparable criteria for defining the success and failure of endodontic treatment; and were published in peer-reviewed medical scientific journals and available in English. These criteria were chosen to ensure that the meta-analysis synthesised relevant and high-quality data, allowing for a thorough analysis of the impact of separated instruments on the success rates of retreatment.
The justification for conducting this systematic review stems from the fundamental need to enhance the predictability and success of endodontic retreatment procedures. By providing valuable insights, this review aims to contribute to the development of more effective treatment strategies.
- The search for studies and statistical treatment and results have no comments.
Answer
I am pleased to note that the search for studies, the statistical treatment and the results sections met with your approval and did not require further comments. Your insights were invaluable in refining our manuscript, and we are grateful for the time and effort you invested in reviewing our work
- Authors must avoid using the first person plural (We) throughout the text. The first sentence of the discussion can be removed (We conducted a systematic review with a meta-analysis to determine the success and failure rates of endodontically retreated teeth that contained a separated instrument prior to retreatment).
ASWER
removed as required (We)
- In this type of study, discussing endodontic success and failure, it is essential to characterize the criteria that are currently adopted regarding success and failure, in addition to addressing questionable cases and establishing a critical analysis of the accuracy between periapical radiography and CBCT. These criteria must be discussed within a current context, involving imaging examination methods with cone beam computed tomography.
ANSWER
I have added to the discussion the considerations relating to periapical radiographs and CBCT as evaluation parameters in the success of endodontic retreatments
Periapical radiography, although widely used, has limitations in sensitivity and specificity, particularly in the detection of periapical lesions. Cone beam computed tomography (CBCT) has become a critical tool due to its increased accuracy in identifying and characterizing periapical pathology. CBCT provides three-dimensional images, allowing for more precise evaluation of the root canal system and surrounding bony structures (2).
Questionable cases, such as those with ambiguous radiographic findings or conflicting clinical symptoms, require thorough evaluation using CBCT to ensure accurate diagnosis and treatment planning. Critical analysis of CBCT compared to periapical radiography shows that although CBCT is superior in terms of diagnostic accuracy, it should be used judiciously considering the higher radiation dose compared to traditional radiographs (3).
The integration of CBCT into endodontic practice improves the ability to evaluate the real extent of the pathology, thus improving the prognostic evaluation of endodontically treated teeth (3).
- Make sure all the text follows the style of the journal. Correct references 7, 12, and 20.
ANSwer
references have been corrected as required
The study is interesting and contributes to the field, but implementing these observations can make it more appealing to readers.
- McGuigan MB, Louca C, Duncan HF. Endodontic instrument fracture: causes and prevention. British dental journal. 2013;214(7):341-8.
- Balasundaram A, Shah P, Hoen MM, Wheater MA, Bringas JS, Gartner A, et al. Comparison of cone-beam computed tomography and periapical radiography in predicting treatment decision for periapical lesions: a clinical study. International journal of dentistry. 2012;2012:920815.
- Ríos-Osorio N, Quijano-Guauque S, Briñez-Rodríguez S, Velasco-Flechas G, Muñoz-Solís A, Chávez C, et al. Cone-beam computed tomography in endodontics: from the specific technical considerations of acquisition parameters and interpretation to advanced clinical applications. Restorative dentistry & endodontics. 2024;49(1):e1.
